# Non-equivalent antigen presenting capabilities of dendritic cells and macrophages in generating brain-infiltrating CD8+ T cell responses

Courtney S. Malo [iD] [1,2], Matthew A. Huggins[1,2], Emma N. Goddery[1,2], Heather M.A. Tolcher[1],
Danielle N. Renner[2,3], Fang Jin[1], Michael J. Hansen[1], Larry R. Pease[1], Kevin D. Pavelko[1] & Aaron J. Johnson[1,4]

The contribution of antigen-presenting cell (APC) types in generating CD8+ T cell responses in the central nervous system (CNS) is not fully defined, limiting the development of vaccines and understanding of immune-mediated neuropathology. Here, we generate a transgenic mouse that enables cell-specific deletion of the H-2Kb MHC class I molecule. By deleting H-2K$^b$ on dendritic cells and macrophages, we compare the effect of each APC in three distinct models of neuroinflammation: picornavirus infection, experimental cerebral malaria, and a syngeneic glioma. Dendritic cells and macrophages both activate CD8+ T cell responses in response to these CNS immunological challenges. However, the extent to which each of these APCs contributes to CD8+ T cell priming varies. These findings reveal distinct functions for dendritic cells and macrophages in generating CD8+ T cell responses to neurological disease.

[1] Department of Immunology, Mayo Clinic, 200 First St SW, Rochester, MN 55905, USA. [2] Mayo Clinic Graduate School of Biomedical Sciences Mayo Clinic, 200 First St SW, Rochester, MN 55905, USA. [3] Neurobiology of Disease Graduate Program Mayo Clinic, 200 First St SW, Rochester, MN 55905, USA. [4] Department of Neurology, Mayo Clinic, 200 First St SW, Rochester, MN 55905, USA. Correspondence and requests for materials should be addressed to A.J.J. (email: Johnson.Aaron2@mayo.edu)

Generating an effective adaptive immune response in the central nervous system (CNS) is a critical goal for treatment of neurotropic pathogens and CNS cancers[1–7]. In particular, the initial activation of pathogen or tumor antigen-specific CD8[+] T cells, and the subsequent entry of these cells into the CNS through a tightly controlled blood brain barrier, is a critical step in this process. Although adaptive immunity in peripheral organs has been studied rigorously, the immune response in the CNS is less characterized. Historically, this is owing to a view that the CNS is immune-privileged[8–10]; however, the CNS is now understood to be immune-specialized rather than immune-isolated[8]. Immune cells, including CD8[+] T cells, regularly enter the CNS in response to pathogens and tumors, and this infiltration is required for protective immunity[3–6,9,11,12]. However, which antigen-presenting cell (APC) type(s) is required for generation of antigen-specific CD8[+] T-cell responses in the CNS, and the location in which they exert their effects, is unclear[13–16].

CD8[+] T cells recognize peptides loaded on specific MHC I molecules, which, in combination with costimulation, results in T-cell receptor signaling and activation and expansion[17]. MHC I molecules are almost ubiquitously expressed, and multiple cell types, including dendritic cells and macrophages, are capable of antigen presentation[13–16,18–21]. Although both of these cell types activate CD8[+] T cells in vitro and peripherally in vivo, whether a response in the CNS is generated through a similar process is unknown[22,23]. As the CNS is distinct from other peripheral tissues, it is imperative to know the contribution of individual APC types. Such knowledge would help optimize CD8[+] T-cell-based immunotherapies for the brain. Likewise, an enhanced understanding of T-cell activation in response to CNS pathogens could lead to novel therapies that reduce autoimmune or pathogen-induced neuropathology.

Previous studies have addressed the role of candidate APCs in response to CNS-derived antigens and the location in which this occurs, including regional lymph nodes[24]. Circulating dendritic cells and macrophages have been demonstrated to be capable of antigen presentation; however, these results were acquired through adoptive transfer techniques or complete ablation of entire cellular subsets, thereby affecting other critical cell functions[15,18,19,25]. To address the specific role of MHC I antigen presentation while leaving all APC subsets intact, we here generate a transgenic mouse that enables conditional deletion of the H-2K[b] (K[b]) MHC I molecule using a cre-lox system. This mouse is devoid of competing endogenous MHC I molecules. We employ this transgenic K[b] LoxP mouse to determine the relative contributions of dendritic cells and macrophages to prime a CD8[+] T-cell response in three distinct models of neuroinflammation. We challenge mice with *Plasmodium berghei* ANKA, Theiler's murine encephalomyelitis virus (TMEV), and GL261 gliomas to examine differences in CD8[+] T-cell responses in each model as a result of conditional MHC I deletion. Here we show a non-redundant role for MHC I antigen presentation by dendritic cells and macrophages in these model systems.

## Results

**H-2K[b] is efficiently deleted in a cell-specific manner.** We generated transgenic K[b] LoxP mice through incorporating LoxP sites that flank the leader sequence of the K[b] gene (Fig. 1a). This animal was generated on a C57BL/6 background and then backcrossed onto a K[b−/−] D[b−/−] background, leaving transgenically expressed K[b] LoxP class I molecule as the sole source of antigen presentation to CD8[+] T cells. We compared expression of K[b] by the inserted transgene in K[b] LoxP mice to levels observed in wild-type C57BL/6 mice. We found no difference in K[b]

expression in cells isolated from the thymus or spleen of K[b] LoxP mice and wild-type C57BL/6 mice (Fig. 1b–d, Supplementary Figure 1). We then crossed the K[b] LoxP mouse to MHC I deficient animals expressing cre recombinase under the CD11c promoter (dendritic cell-specific) or LysM promoter (monocyte/granulocyte/macrophage-specific)[26,27]. This generated CD11c-cre K[b] conditional knockout (cKO) and LysM-cre K[b] cKO animals (Fig. 1e). We observed efficient deletion of K[b] on dendritic cells and macrophages in the spleen of CD11c-cre K[b] cKO and LysM-cre K[b] cKO animals, respectively, demonstrating effective cell-specific silencing (Fig. 1f–i). This system left activated CD11c[+] microglia largely unaffected, as demonstrated by K[b] expression (Supplementary Figure 2). Additionally, we generated cytomegalovirus promoter (CMV)-cre K[b] cKO mice, which ubiquitously express cre, resulting in full K[b] deletion (Fig. 1f, g). These experiments demonstrate that the transgenic K[b] cKO mouse model system is functioning as expected and is appropriate to address the contribution of specific APCs in development and following antigenic challenge.

**T-cell development is unaffected by APC-specific K[b] deletion.** To determine the extent conditional ablation of K[b] affected T-cell development, we evaluated the frequency of CD4 and CD8 single positive thymocytes in naïve CD11c-cre K[b] cKO, LysM-cre K[b] cKO, and CMV-cre K[b] cKO animals. Cre-negative littermates served as negative controls. As shown in Fig. 2a–d, proportions of CD4 and CD8 single-positive thymocytes are unaffected by CD11c-cre or LysM-cre-mediated K[b] deletion (Fig. 2a–d). In contrast, CMV-cre K[b] cKO mice present with dysregulated thymic selection from double-positive thymocytes to the CD8 single-positive thymocyte pool (Fig. 2b). This is expected, as there is no positive selection of CD8[+] T cells in the thymus when K[b] is deleted[18]. To further assess if CD11c-cre or LysM-cre-mediated deletion of K[b] impacted CD8[+] T-cell development, we also measured proportions of immune cells in the spleen. As shown in Fig. 2e–h, we found no difference in the proportion of CD4[+] and CD8[+] T cells in the spleen of naïve animals, despite efficient K[b] deletion in each APC cell type. These findings demonstrate that CD8[+] T-cell development is unaltered by conditional deletion of K[b] on CD11c[+] dendritic cells or LysM[+] macrophages.

To determine if conditional deletion of MHC I on APC impacted T-cell repertoire diversity, we analyzed TCR Vβ usage in naïve animals. We did not observe marked differences in CD11c-cre K[b] cKO and LysM-cre K[b] cKO compared to littermate controls. A minor 3% reduction was found in TCR Vβ5 usage by CD8[+] T cells (Fig. 3a). This slight reduction in TCR Vβ 5 usage was also observed in CD8 single-positive thymocytes (Supplementary Figure 3). Interestingly, CD8 single-positive thymocytes also exhibited some minor variation in TCR Vβ 8 usage among the three genotypes (Supplementary Figure 3). However, this difference was not observed in the mature CD8[+] T-cell repertoire (Fig. 3a). The usage of all other Vβ chains was comparable between K[b] cKO mice and littermate controls. Therefore, conditional deletion of H-2K[b] was not overtly affecting the mature CD8[+] T-cell repertoire.

Moreover, CD11c-cre K[b] cKO, LysM-cre K[b] cKO, and CMV-cre K[b] cKO mice do not present with symptoms of immune dysregulation when following animals beyond one year of age. Therefore, we do not observe notable effects of APC-specific MHC I deletion on thymic development or evidence of autoimmunity, making these animals suitable to analyze antigen presentation following direct immunologic challenge in our model systems.

**Macrophage-specific K[b] deletion does not protect against ECM.** We next addressed the requirement of K[b] on dendritic cells and

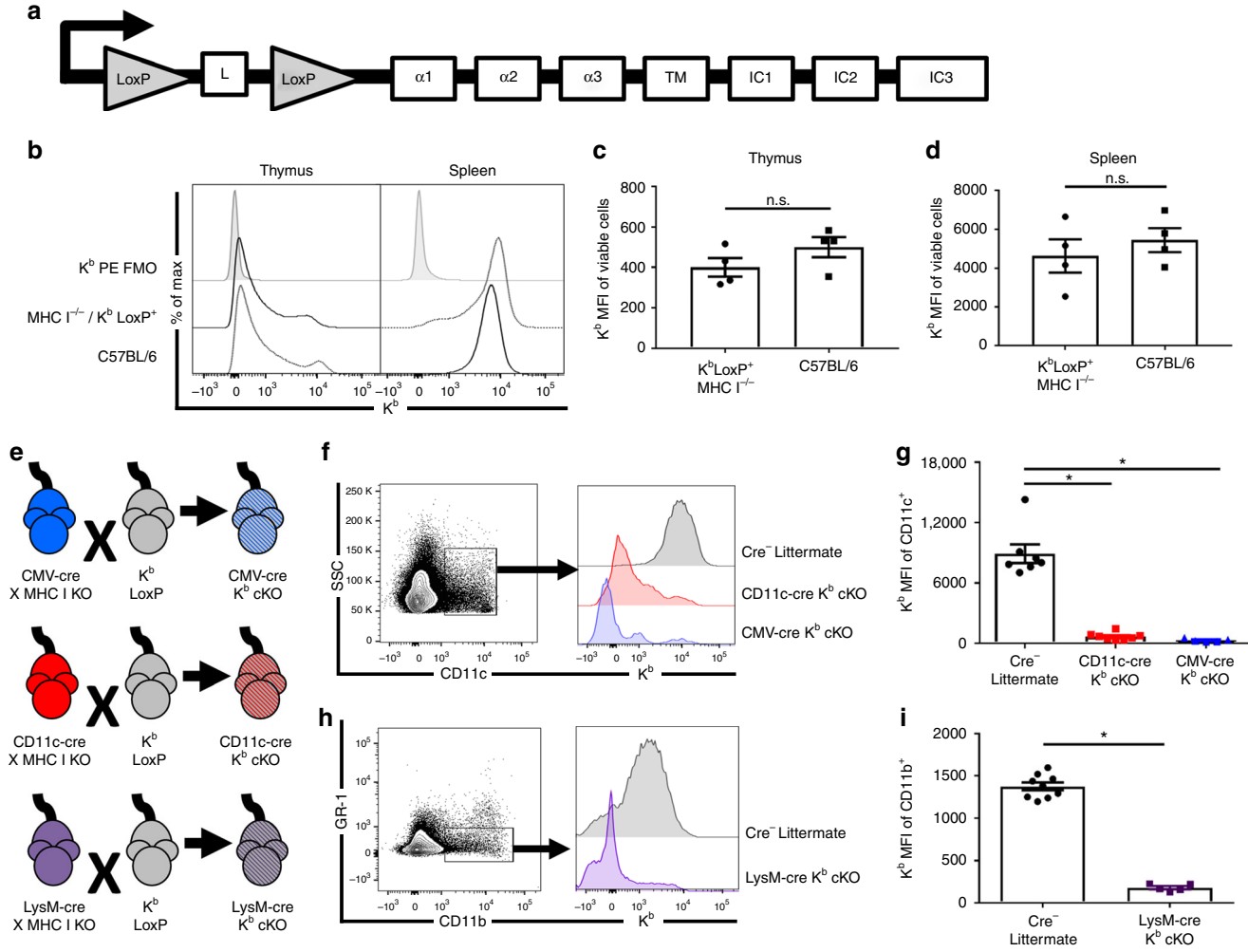

**Fig. 1** Normal protein expression and efficient cell-specific deletion in H-2K$^b$ LoxP transgenic mice. **a** Diagram of the K$^b$ transgene in which the leader sequence (L) is flanked by LoxP sites (TM: transmembrane, IC: intracellular). **b** Representative histograms and **c** and **d** quantification of median fluorescence intensity (MFI) of K$^b$ on total live cells in the thymus and spleen demonstrate no difference in expression of K$^b$ derived from the transgene ($N = 4$ per group). **e** Schematic of K$^b$ cKO crosses. **f** Representative histogram and **g** median fluorescence intensity (MFI) quantification showing efficient knockdown of K$^b$ on CD11c$^+$ dendritic cells ($N = 7$, 8, 5 per group, respectively). **h** Representative histogram and **i** MFI demonstrate K$^b$ knockdown on CD11b$^+$ GR-1$^-$ macrophages ($N = 8$ and 7 per group, respectively). Data presented as mean with error bars representing standard error of the mean (SEM). * denotes $p < 0.05$ by Mann−Whitney $U$ Test or one-way ANOVA with Holm−Sidak correction

macrophages to prime CD8$^+$ T-cell responses in the *P. berghei* ANKA infection model of experimental cerebral malaria (ECM). Infection of immunocompetent C57BL/6 animals with *P. berghei* ANKA results in extensive blood−brain barrier disruption mediated by CD8$^+$ T cells 6 days following inoculation with parasitized red blood cells (Fig. 4a)[11,28]. Animals typically become moribund and succumb to disease at this time point. We found that CD11c-cre and LysM-cre K$^b$ cKO animals both had reduced infiltration of CD8$^+$ T cells into the CNS 6 days post infection (Fig. 4e–g). This effect was not due to a difference in infiltrating CD45$^{hi}$ immune cells or specifically CD4$^+$ T cells (Fig. 4c, d). These results suggest a requirement for K$^b$-restricted antigen presentation by both macrophages and dendritic cells. Notably, there was also no difference in circulating parasite load between CD11c-cre K$^b$ cKO, LysM-cre K$^b$ cKO, and cre-negative animals (Fig. 4h).

Interestingly, a differential requirement of class I expression on dendritic cells and macrophages was observed pertaining to the onset of neuropathology. We therefore assessed vascular permeability by administering fluorescein isothiocyanate (FITC) conjugated to albumin protein intravenously 6 days post infection

(Fig. 5a). FITC-albumin cannot cross the blood−brain barrier under normal conditions. Therefore, increased FITC-albumin in the parenchyma of the brain is the result of blood−brain barrier disruption, a hallmark of lethal *P. berghei* ANKA infection. We observed using confocal microscopy that the blood−brain barrier was disrupted in cre-negative littermate controls and LysM-cre K$^b$ cKO mice despite a reduction in CD8$^+$ T-cell infiltration in LysM-cre K$^b$ cKO mice (Fig. 5b). Conversely, CD11c-cre K$^b$ ckO and CMV-cre K$^b$ ckO animals maintained organization of blood vessels in the CNS, as shown by FITC-albumin retention in the vessel (Fig. 5b). In addition, occludin staining for both CMV-cre K$^b$ cKO and CD11c-cre K$^b$ cKO samples demonstrate a linear staining pattern, which is not present in LysM-cre K$^b$ cKO or cre-negative littermate samples (Fig. 5b). We also quantified CNS vascular permeability by assessing fluorescence of whole brain homogenates following FITC-albumin injection. We determined that CD11c-cre and CMV-cre K$^b$ cKO animals maintained vascular integrity after *P. berghei* ANKA infection. Conversely, we found extensive FITC-albumin leakage in LysM-cre K$^b$ cKO animals, despite these animals exhibiting reduced infiltration of CD8$^+$ T cells (Fig. 5c). Consistent with their extent of blood

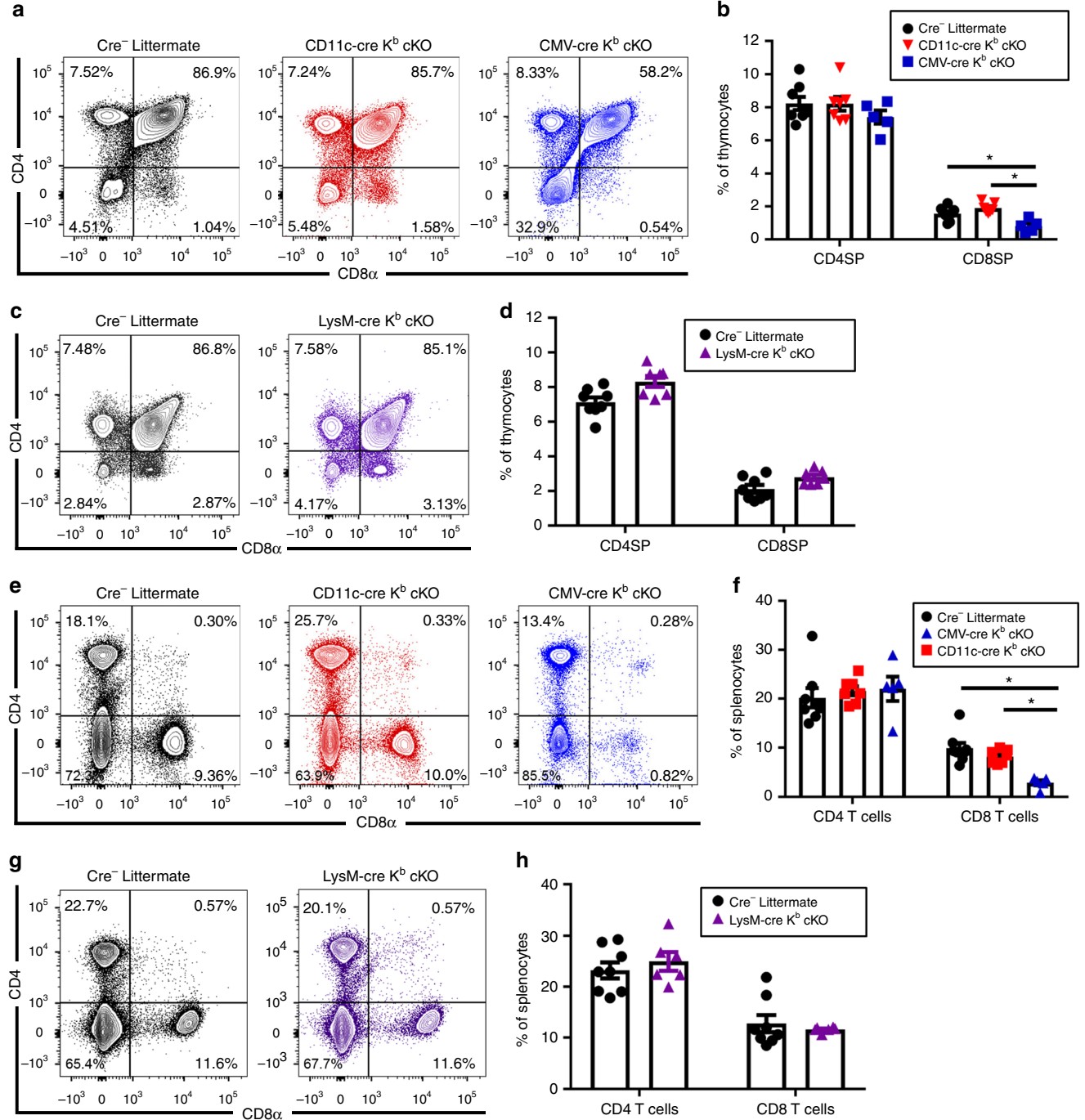

**Fig. 2** Normal T-cell development in naive CD11c-cre and LysM-cre K[b] cKO mice. **a** Representative plots and **b** quantification show a block in thymocyte development only in CMV-cre K[b] cKO animals, but not CD11c-cre K[b] cKO animals ($N = 7, 8, 5$ per group, respectively). **c** Representative plots and **d** quantification show no disruption of thymocyte development in LysM-cre K[b] cKO animals ($N = 8$ and 7 per group, respectively). **e** Representative plots and **f** quantification show abnormal splenic CD8[+] T-cell proportions in naïve CMV-cre K[b] cKO animals, but not CD11c-cre K[b] cKO animals. **g** Representative plots and **h** quantification show normal splenic T-cell populations in naïve LysM-cre K[b] cKO animals. Data presented as mean with error bars representing standard error of the mean (SEM). * denotes $p < 0.05$ by Mann−Whitney $U$ Test or one-way ANOVA with Holm−Sidak correction

−brain barrier permeability, CD11c-cre K[b] cKO animals were fully protected from ECM with 100% of animals surviving (Fig. 5d). Conversely, LysM-cre K[b] cKO and cre-negative littermate controls succumbed to ECM between days 6 and 7 post challenge, similar to wild-type C57BL/6 mice (Fig. 5d). These results demonstrate a role for both dendritic cells and macrophages in generating a CD8[+] T-cell response to *P. berghei* ANKA infection. However, only antigen presentation by dendritic cells

results in a CD8[+] T-cell response capable of inducing fatal blood −brain barrier disruption as a result of *P. berghei* ANKA infection.

**K[b] deletion impairs CD8[+] T-cell responses to TMEV infection.** We next assessed the capacity for CD8[+] T-cell priming during acute brain infection with recombinant TMEV encoding the

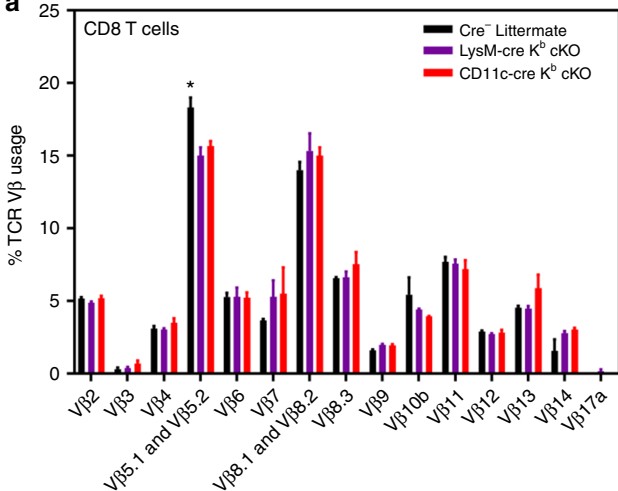

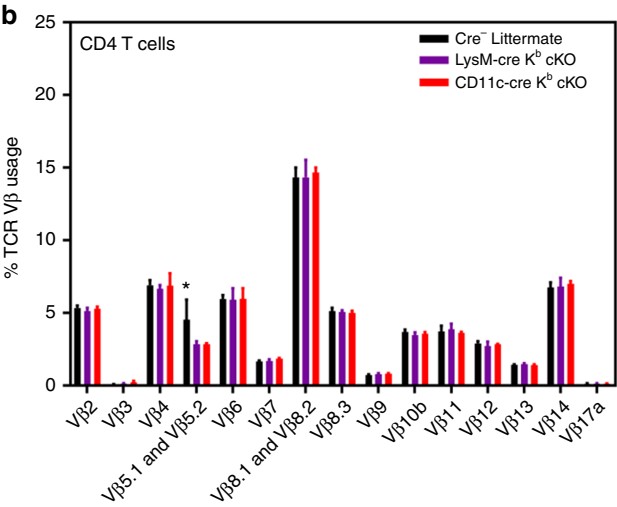

**Fig. 3** Splenic TCR repertoire is not markedly affected by cell-specific H-2K$^b$ deletion. Splenocytes were isolated from naïve 6–8-week-old animals ($N$ = 3 per group) and stained with anti-CD45, anti-CD4, and anti-CD8 antibodies, as well as antibodies specific to each of the T-cell receptor Vβ regions listed. CD8$^+$ T cells (**a**) and CD4$^+$ T cells (**b**) isolated from the spleen show highly similar T-cell repertoires, with TCR Vβ5 only impacted peripherally. Data presented as mean with error bars representing standard error of the mean (SEM). * denotes $p < 0.05$ as measured by two-way ANOVA with Holm−Sidak correction

model K$^b$-restricted OVA (SIINFEKL) peptide[29]. Intracranial infection with TMEV-OVA generates a robust CD8$^+$ T-cell response to OVA peptide presented in the context of the K$^b$ MHC I molecule 7 days post infection[5,29]. We intracranially infected CD11c-cre K$^b$ cKO, CMV-cre K$^b$ cKO, LysM-cre K$^b$ cKO, and cre-negative littermate controls with TMEV-OVA and isolated brain infiltrating lymphocytes 7 days post infection (Fig. 6a). We demonstrated a significant reduction in the proportion of K$^b$: OVA-specific CD8$^+$ T cells in the CNS of CD11c-cre K$^b$ cKO and CMV-cre K$^b$ cKO animals (Fig. 6f). However, LysM-cre K$^b$ cKO animals were able to mount K$^b$:OVA-specific antiviral CD8$^+$ T-cell responses similar to cre-negative littermate controls (Fig. 6f). Overall CD45$^{hi}$ infiltration, as well as infiltration of CD4$^+$ T cells, in the CNS were not changed by cell-specific MHC I deletion (Fig. 6c, d). This was not the result of differences in viral load in the CNS 7 days post infection (Supplementary Figure 4)[30]. This demonstrates that K$^b$-expressing dendritic cells are required to present OVA peptide to

CD8$^+$ T cells to generate an antigen-specific CD8$^+$ T-cell response. Meanwhile, macrophages were not responsible for priming CD8$^+$ T-cell responses to CNS viral antigens in response to TMEV-OVA infection.

**K$^b$ expression by DCs is required for anti-tumor immunity.** We next assessed the CD8$^+$ T-cell response generated against CNS tumors using the GL261 syngeneic glioma model. GL261 gliomas can be inoculated in immune-competent C57BL/6 mice, allowing for the study of adaptive immunotherapy[5–7,31,32]. A modified GL261 glioma line, termed GL261-quad cassette, expresses OVA peptide and enables analysis of K$^b$:OVA-specific CD8$^+$ T-cell responses[5–7]. We observed little effect of K$^b$ deletion on the naturally generated immune response to GL261-quad gliomas 21 days post tumor inoculation. This is expected, as there is very little response to this glioma in unvaccinated animals (Supplementary Figure 5). This suggests that in order to assess the role of dendritic cells and macrophages in generating a CD8$^+$ T-cell response to GL261-quad cassette gliomas, a vaccination strategy would be necessary. Our group has previously demonstrated a beneficial effect of boosting K$^b$:OVA-specific CD8$^+$ T cells through TMEV-OVA vaccination in this model[6,7,29,33]. Using this established vaccination protocol, we challenged CD11c-cre K$^b$ cKO, CMV-cre K$^b$ cKO, LysM-cre K$^b$ cKO, and cre-negative littermate control animals with GL261-quad gliomas and vaccinated with TMEV-OVA intracranially (Fig. 7a). Two weeks following TMEV-OVA vaccination, we isolated brain-infiltrating immune cells from animals of each genotype. We found that CD11c-cre K$^b$ cKO animals had reduced tumor antigen-specific CD8$^+$ T cells compared to littermate controls. However, this reduction was not as complete as that observed in vaccinated CMV-cre K$^b$ cKO mice, which had a complete loss of responding K$^b$:OVA epitope-specific CD8$^+$ T cells (Fig. 7e, Supplementary Figure 5). There was no difference in infiltration of immune cells or total CD8$^+$ T cells (Fig. 7c, d). Conversely, LysM-cre K$^b$ cKO had comparable K$^b$:OVA-specific CD8$^+$ T-cell responses to cre-negative littermate controls (Fig. 7f–i).

We also assessed tumor size weekly in CD11c-cre K$^b$ cKO, CMV-cre K$^b$ cKO, LysM-cre K$^b$ cKO, and cre-negative littermate control mice. In agreement with our assessment of brain-infiltrating, K$^b$:OVA-specific CD8$^+$ T cells, CMV-cre K$^b$ cKO animals had extensive tumor growth (Fig. 8a, b). Also in agreement with our analysis of CD8$^+$ T-cell responses, CD11c-cre K$^b$ cKO animals bearing GL261-quad gliomas displayed intermediate control of tumor burden, mirroring an intermediate immune response (Fig. 8a, b). Finally, we observed no difference in GL261-quad tumor size in LysM-cre K$^b$ cKO compared to littermate controls. This implied that similar to acute TMEV-OVA infection, there was not a role for macrophages in generating CD8$^+$ T-cell responses specific to glioma antigens during this vaccination regimen (Fig. 8c, d). Meanwhile, CD11c$^+$ dendritic cells were the major APC contributing to CD8$^+$ T-cell priming against tumor in the GL261-Quad model.

**Discussion**
Macrophages, dendritic cells, and other CNS-resident cells have each been reported to prime antigen-specific CD8$^+$ T-cell responses[1–4,18,32,34,35]. However, the relative contribution of each of these cell types was previously undefined. In this study, we have demonstrated a critical role of dendritic cells in generating CD8$^+$ T-cell responses in three distinct models of neuroinflammation. We have additionally shown that macrophages contribute to antigen presentation. CD11c$^+$ dendritic cells play a major role in antigen presentation to CD8$^+$ T cells in the context of a CNS viral infection, immunopathology, and vaccination

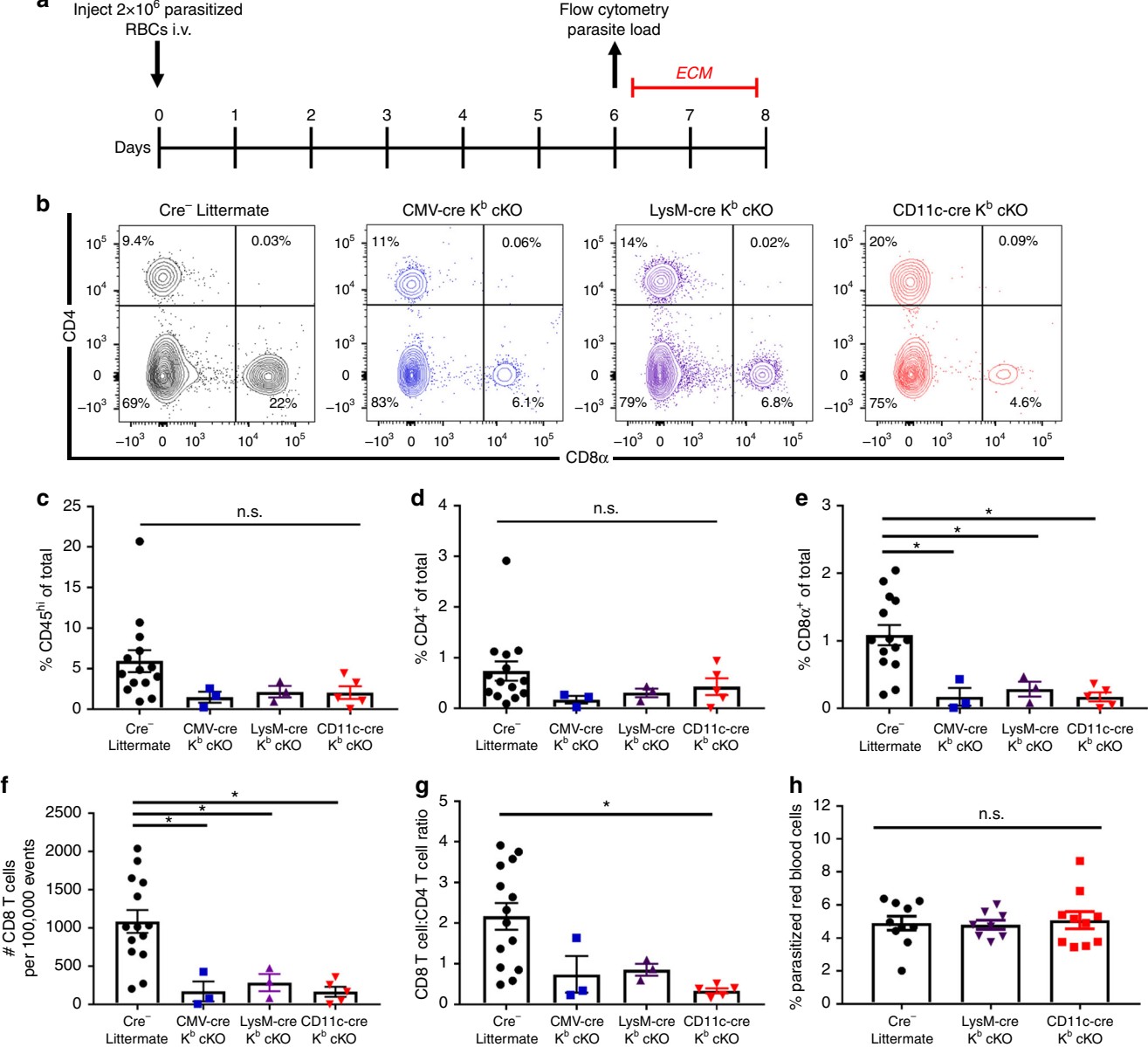

**Fig. 4** Dendritic cells and macrophages generate discrete CD8[+] T-cell responses in cerebral malaria. **a** Timeline of infection and readout with *Plasmodium berghei* ANKA. **b** Representative flow plots and **c**−**g** quantification demonstrate a reduction proportion and number of brain-infiltrating CD8[+] T cells, but not CD45[hi] immune cells or CD4[+] T cells (*N* = 14, 3, 3, 5 per group, respectively). **h** No difference in proportion of parasitized red blood cells was found between cre-negative littermates and conditional knockout stains 6 days post infection (*N* = 10, 8, 10 per group, respectively). Data presented as mean with error bars representing standard error of the mean (SEM). * denotes *p* < 0.05 by Mann−Whitney *U* Test or one-way ANOVA with Holm−Sidak correction

against a CNS tumor. LysM[+] macrophages contribute to parasite-specific CD8[+] T-cell priming, but LysM[+] macrophages were not responsible for productive MHC I antigen presentation in the context of viral infection or CNS tumors. These results demonstrate nonredundant roles for macrophages and dendritic cells in priming CD8[+] T-cell responses in the CNS, with the latter playing a more predominant role in these three neuropathology models.

Due to the potential requirement for antigen presentation by hematopoietic cells during thymic development, we initially investigated T-cell development in the thymus and spleen[36,37]. We observed no difference in CD8[+] T-cell development despite conditional deletion of K[b] on dendritic cells or macrophages. Specifically, thymocyte or splenocyte populations were comparable to cre-negative littermate controls. This indicates, from a

population-based perspective, that there is not a requirement for antigen presentation by dendritic cells or macrophages for CD8[+] T-cell development in the thymus or homeostatic maintenance in the spleen[38,39]. This stands in contrast to previous reports in which dendritic cells were completely ablated, resulting in thymic atrophy[40]. This result was likely due to functions lost by ablation of dendritic cells that were not related to MHC I. Conversely, our positive control CMV-cre K[b] cKO animals, which ubiquitously lose K[b] expression, displayed disrupted CD8[+] T-cell development. This was expected, as MHC I is required for positive selection of CD8 single-positive thymocytes[22]. In addition, we observed a small reduction in CD8[+] T cells utilizing TCR Vβ 5, implying a minor change in the TCR repertoire. However, this reduction was modest, accounting for only a 3% difference in our analysis.

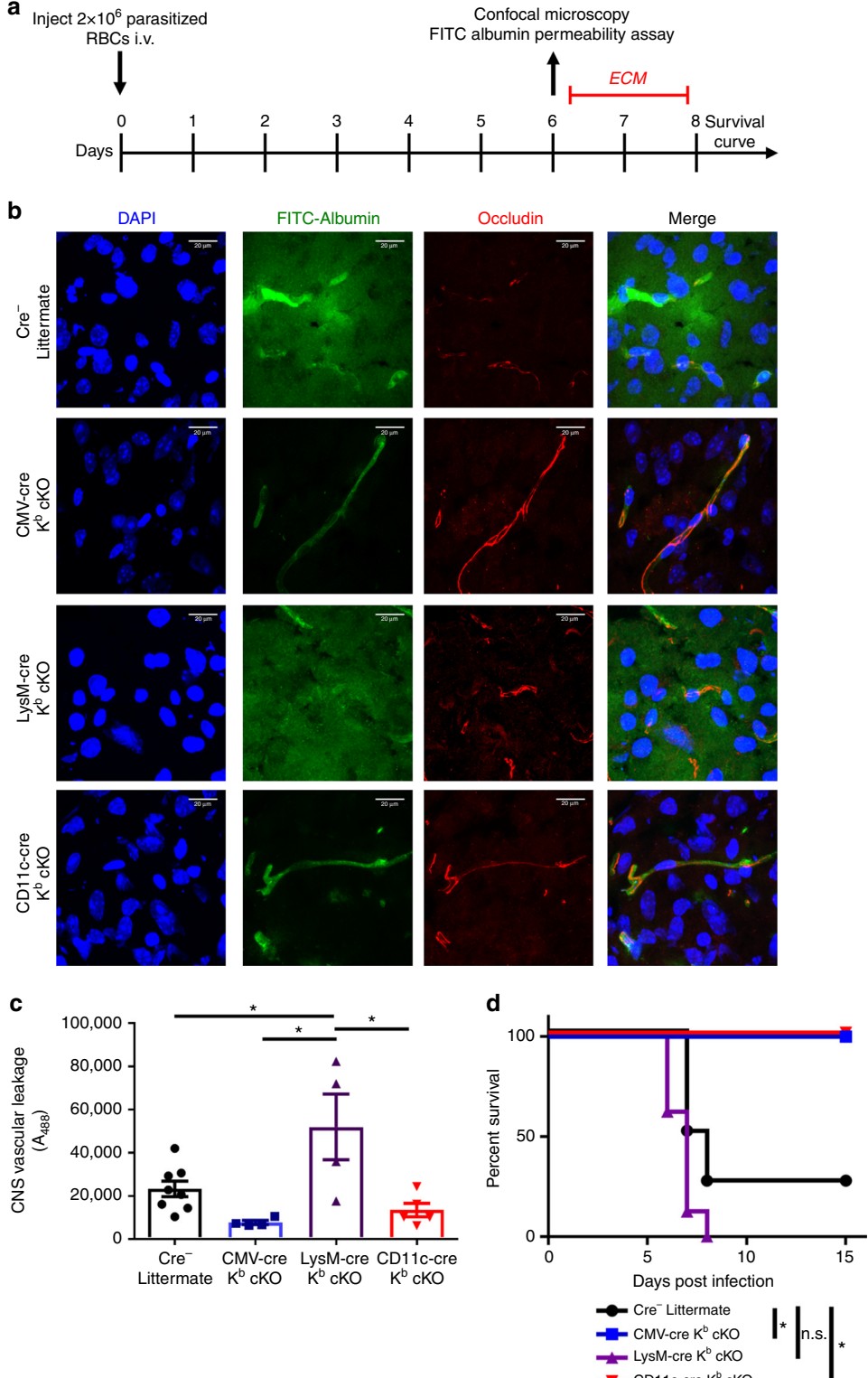

**Fig. 5** Dendritic cell-specific deletion of $K^b$ protects mice from lethal blood−brain barrier disruption. **a** Timeline of infection with *Plasmodium berghei* ANKA. **b** Representative microscopy images demonstrate blood−brain barrier disruption in cre-negative littermates and LysM-cre $K^b$ cKO animals, but not CD11c-cre or CMV-cre $K^b$ cKO animals. Scale bar represents 20 μm. **c** Fluorescence of whole brain homogenate demonstrates extensive leakage of FITC-albumin in cre-negative littermates and LysM-cre $K^b$ cKO animals ($N = 8, 4, 4, 12$ per group, respectively). **d** Dendritic cell-specific, but not macrophage-specific deletion of $K^b$ provided a survival advantage ($N = 8, 5, 8, 12$ per group, respectively). Data presented as mean ± SEM. * denotes $p < 0.05$ by Mann−Whitney *U* Test or one-way ANOVA with Holm−Sidak correction

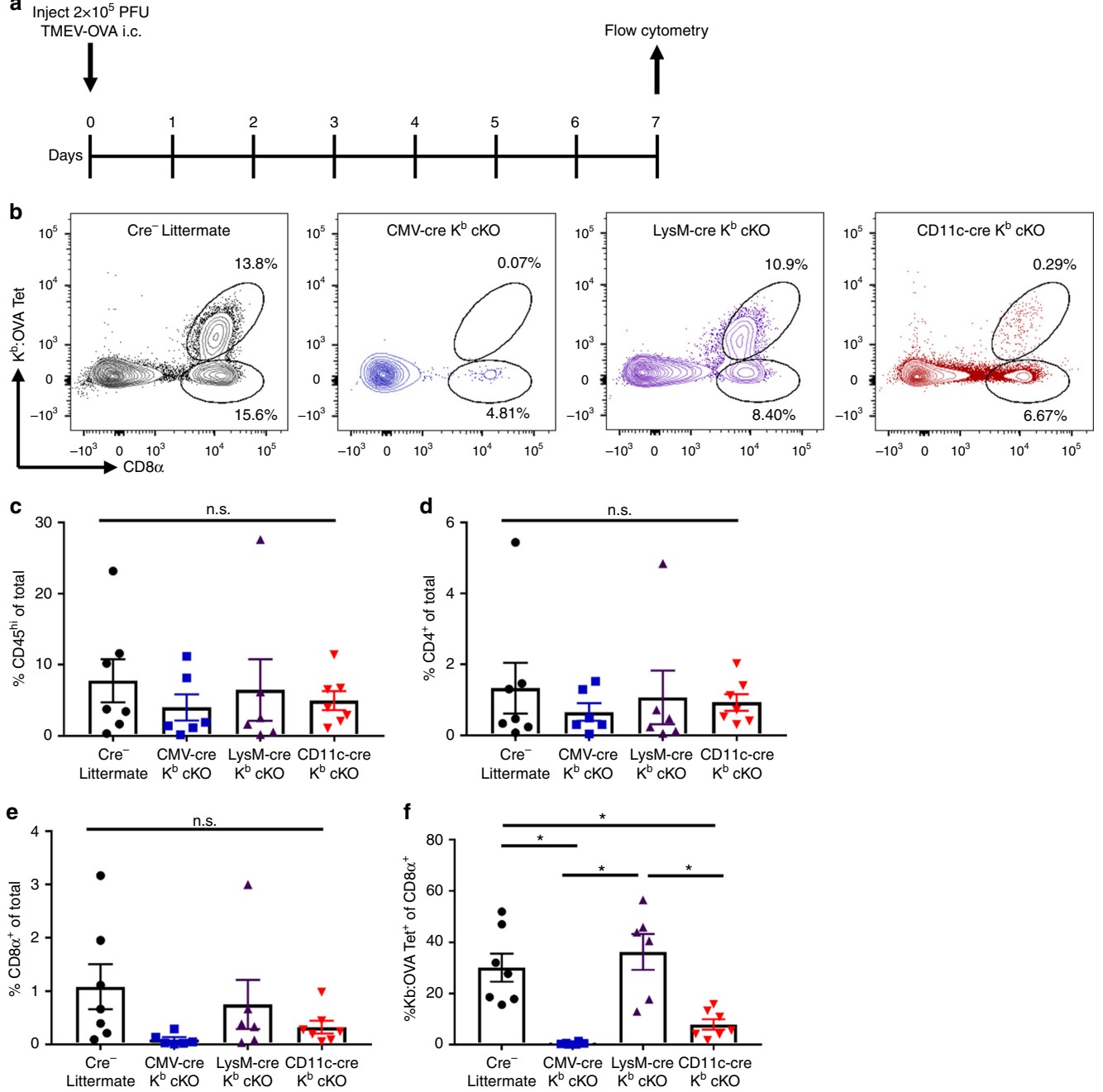

**Fig. 6** Dendritic cell-specific deletion of K$^b$ ablates CD8$^+$ T-cell response to picornavirus infection. **a** Animals were challenged with $2 \times 10^5$ plaque forming units (PFU) TMEV-OVA. Brains were isolated for flow cytometry 7 days post infection. **b** Representative flow plots. **c** Quantification of CD45$^{hi}$ (**c**) CD4$^+$ T cells (**d**) or total CD8$^+$ T cells (**e**) demonstrate no difference in immune cell infiltration into the CNS. **f** Quantification demonstrates that CD11c-cre K$^b$ cKO animals are unable to generate CD8$^+$ T-cell responses following acute CNS infection with TMEV-OVA ($N = 7, 6, 6, 7$ per group, respectively). **d** Quantification of OVA-specific CD8$^+$ T-cell responses were tracked using K$^b$:OVA MHC I tetramer. Data presented as mean with error bars representing standard error of the mean (SEM). * denotes $p < 0.05$ by Mann–Whitney $U$ Test or one-way ANOVA with Holm–Sidak correction

Meanwhile, the usage of all other Vβ chains was consistent across genotypes in mature CD8$^+$ T-cell populations. Therefore, APC-specific deletion of MHC I does not result in substantial immune dysregulation in the absence of immunologic challenge.

We determined that conditional deletion of MHC I on either dendritic cells or macrophages resulted in reduced frequency of brain-infiltrating CD8$^+$ T cells in response to *P. berghei* ANKA infection. Given these results, we expected both dendritic cell- and macrophage-specific deletion of MHC I would provide a survival advantage compared to cre-negative littermates. However, this was not observed. Deleting K$^b$ on dendritic cells fully protects animals from ECM, whereas deletion of K$^b$ on macrophages still resulted in CD8$^+$ T-cell- induced blood–brain barrier disruption and conferred no survival advantage. Recently, we reported that adoptively transferred, perforin competent CD8$^+$ T cells contribute to fatal brain edema in ECM[11]. This implies that there may

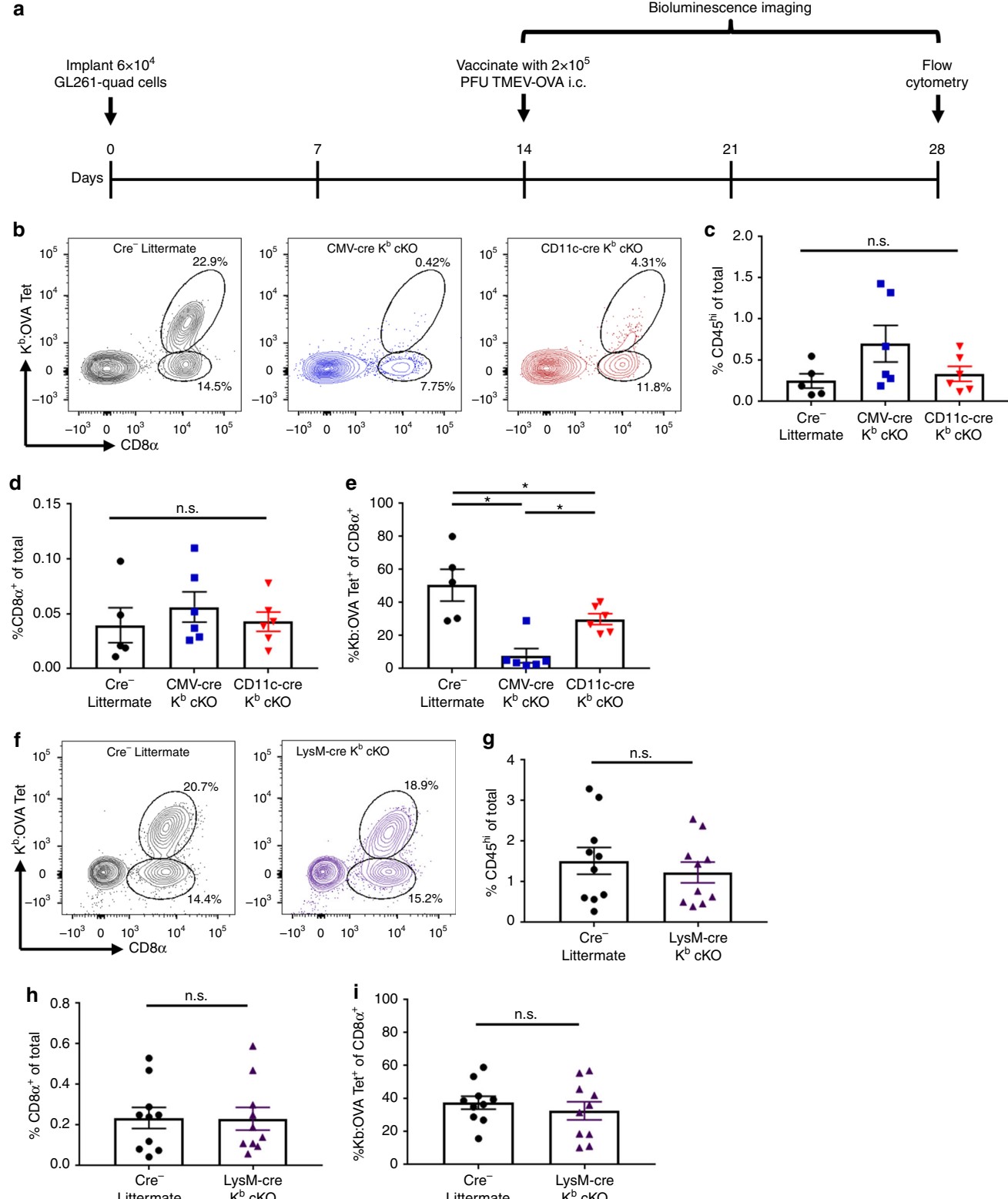

**Fig. 7** Dendritic cell-specific deletion of $K^b$ disrupts CD8+ T-cell priming against gliomas. **a** Timeline of GL261-quad cassette tumor inoculation, bioluminescence imaging, and vaccination. **b** Representative flow cytometry plots. There was no difference in CD45$^{hi}$ cell (**c**) or CD8+ T-cell (**d**) infiltration into the CNS of GL261-bearing animals. **e** Quantification demonstrate that CD11c-cre $K^b$ cKO animals lose the ability to respond to GL261-gliomas following vaccination, though not as severely as CMV-cre $K^b$ cKO animals ($N = 5, 6, 6$ per group, respectively). **f** Representative plots and **g**–**i** quantification demonstrate that LysM-cre $K^b$ cKO animals respond to GL261-quad cassette gliomas similarly to controls ($N = 10$ per group). Data presented as mean with error bars representing standard error of the mean (SEM). * denotes $p < 0.05$ by Mann–Whitney $U$ Test or one-way ANOVA with Holm–Sidak correction

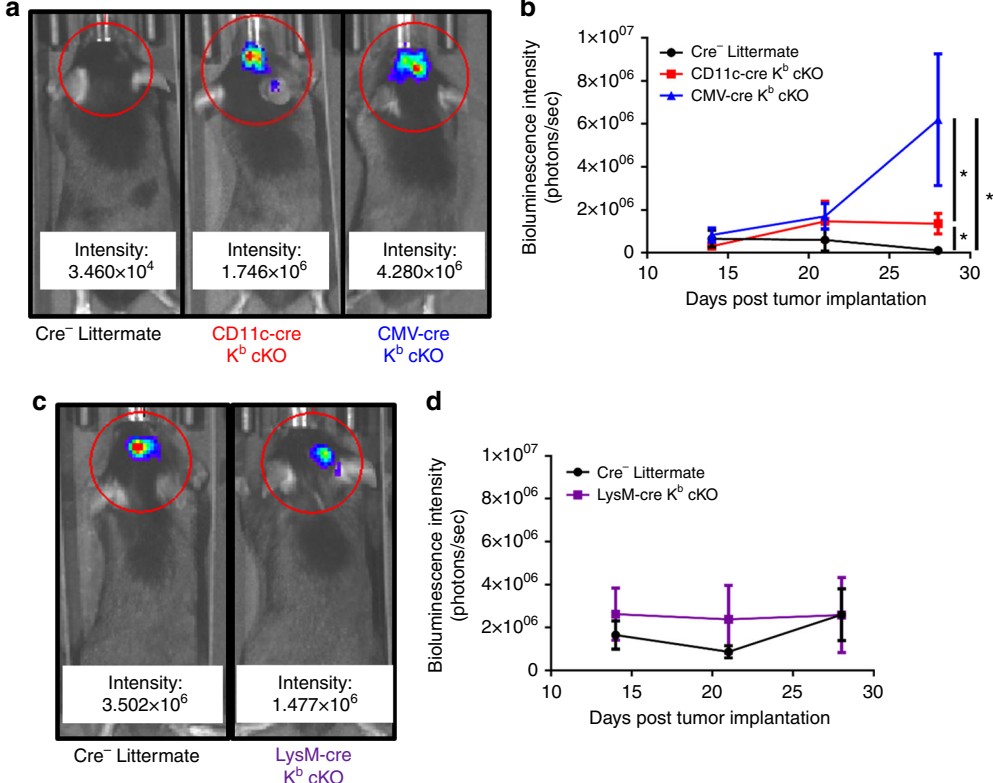

**Fig. 8** CD11c-cre K$^b$ cKO mice have intermediate ability to control GL261 glioma growth. Tumor burden in GL261-quad cassette-bearing animals was assessed weekly via bioluminescence imaging beginning at day 14 post tumor inoculation. **a** Representative images at day 28 and **b** quantification demonstrate a loss of tumor control in CMV-cre K$^b$ cKO animals, and a less severe loss of tumor control in CD11c-cre K$^b$ cKO animals ($N = 5, 6, 6$ per group, respectively). **c** Representative images at day 28 and **d** quantification demonstrate that tumor growth in LysM-cre K$^b$ cKO animals and cre-negative littermates was comparable ($N = 10$ per group). Data presented as mean with error bars representing standard error of the mean (SEM). * denotes $p < 0.05$ by Mann–Whitney $U$ Test or one-way ANOVA with Holm–Sidak correction

be a qualitative difference in CD8$^+$ T cells stimulated by dendritic cells or macrophages in this ECM model. Understanding the difference is potentially important, since it implies that targeting antigen presentation on different cell types could be a therapeutic approach to reduce fatal blood–brain barrier disruption.

In these studies, we also determined that cell-specific K$^b$ deletion impacted the generation of virus-specific CD8$^+$ T-cell responses following intracranial picornavirus infection. We found that dendritic cells, but not macrophages, are required to present OVA antigen to prime the virus-specific CD8$^+$ T-cell response against acute TMEV-OVA infection. Furthermore, dendritic cells appear to be the sole APC type responsible for acute CD8$^+$ T-cell priming, as the K$^b$:OVA-specific CD8$^+$ T-cell response was abolished to levels observed in animals in which K$^b$ was ubiquitously deleted. This suggests that other cell types are unable to compensate for MHC I loss on dendritic cells in the context of an acute picornavirus infection in the CNS. Importantly, this difference observed was not the result of different viral load at 7 days post virus infection. It is important to note that the long-term CD8$^+$ T-cell response to viral infection was not evaluated in these studies and will be the topic of future experiments.

Pertaining to their role in the GL261 glioma model, we found that dendritic cells, but not macrophages, were again the most effective at priming tumor antigen-specific CD8$^+$ T-cell responses in the brain. When we vaccinated animals using TMEV-OVA picornavirus vaccine, we found notable differences in CD8$^+$ T-cell responses between cre-negative littermates and CD11c-cre K$^b$ cKO mice. This again implies a role for dendritic cells, but not macrophages, in generating tumor antigen-specific CD8$^+$ T-cell response. Importantly, unlike our results analyzing virus infection with TMEV-OVA, the reduction in tumor antigen-specific CD8$^+$ T cells in CD11c-cre K$^b$ cKO animals was not as complete as CMV-cre K$^b$ cKO animals. CD11c-cre K$^b$ cKO animals had an intermediate level of CD8$^+$ T-cell priming. This suggests a potential contribution of an additional APC subset which remains undefined. Nevertheless, despite literature suggesting a therapeutic role for macrophages in the context of treating gliomas, our data contend that the primary function of these cells is not through MHC I-restricted priming of anti-tumor CD8$^+$ T-cell responses[41–44]. Importantly, as standard-of-care treatments may modulate efficacy of immunotherapies, determining which APCs are necessary beyond dendritic cells could be important for future design of rational treatment regimens[45,46].

The importance of CD11c$^+$ and LysM$^+$ cells in antigen priming of CD8$^+$ T cells is evident from these studies. However, notable questions remain regarding the role of other candidate APC types and the location of antigen presentation. The trafficking of antigen and cells from the CNS to peripheral lymphoid compartments during CD8$^+$ T-cell priming also needs to be further defined. Importantly, the role of antigen presentation by microglia, astrocytes, and other CNS-resident APC can be addressed employing the K$^b$ LoxP mouse model. With the recent recognition of lymphatics in the brain, the question of where antigen acquisition occurs may begin to be addressed[47]. It is possible that

free antigen is captured in the draining cervical lymph node by APCs[44,48]. Alternatively, antigen could be taken up by APCs in the CNS and carried to the lymph node[21]. Finally, an effort to determine how various APCs affect sustained and memory CD8+ T-cell responses in the CNS needs to be addressed. While this was not addressed in these experiments, these questions are important and will be addressed in subsequent studies. The development of a system to conditionally silence class I molecules in vivo will greatly enhance these future studies.

## Methods

**Generation of transgenic K$^b$ cKO strains**. The H-2K$^b$ LoxP mouse was generated by our group. LoxP sites were inserted into a K$^b$ transgene which was previously cloned using site-directed mutagenesis[12]. The LoxP sites flank the Kozak and leader sequence of the K$^b$ gene. Thus, when cre recombinase is expressed, this gene segment is eliminated, preventing protein translation. The transgene was introduced to C57BL/6 mice by the Mayo Clinic Transgenic Mouse Core (Rochester, MN). These animals were then backcrossed onto a MHC I-deficient background until they lacked endogenous MHC I. Animals were screened for the transgene insertion by expression of K$^b$ via flow cytometry.

Commercially available CD11c-cre (B6.Cg-Tg(Itgax-cre)1-1Reiz/J, 008068), LysM-cre (B6.129P2-Lyz2tm1(cre)Ifo/J, 004781), and CMV-cre (B6.C-Tg(CMV-cre)1Cgn/J, 006054) animals were crossed to MHC I-deficient animals for at least three generations for each strain to be MHC I-deficient (The Jackson Laboratory, Bar Harbor, ME). These animals were crossed to the K$^b$ LoxP mouse to generate each strain of K$^b$ cKO animals. Deletion of K$^b$ was confirmed by polymerase chain reaction for cre using primer sequences available from the Jackson Laboratory as well as flow cytometry for K$^b$ MHC I surface protein expression. All animal experiments were approved by and performed in accordance to the Mayo Clinic Institutional Animal Care and Use Committee.

**Plasmodium berghei ANKA infection**. Red blood cells parasitized with *P. berghei* ANKA were provided by the lab of John T. Harty, Ph.D., Department of Microbiology, University of Iowa, Iowa City, IA. Mice between 8 and 12 weeks old were infected with 10$^6$ parasitized red blood cells intravenously via retro-orbital injection[11]. Mice were anesthetized with 1–2% isoflurane, and then received a single dose of 10$^6$ parasitized red blood cells behind the right eye in a total volume of 100 μL sterile saline. Mice were killed on day 6 post infection and processed for flow cytometry. To address blood–brain barrier disruption, 100 mg/ml FITC (Sigma-Aldrich, St. Louis, MO) in phosphate-buffered saline was injected intravenously via retro-orbital injection one hour before the killing. Brains were subsequently frozen in aluminum foil on dry ice and stored at −80 °C.

**Confocal microscopy**. Fresh frozen tissue from the right hemisphere of FITC-albumin injected animals were embedded in Tissue-Tek OCT compound (Sakura Finetek, Torrance, CA) cut on a cryostat and placed on positively charged slides. Ten micrometers thick coronal sections were cut from the isocortex anterior to the hippocampal formation, which has been previously demonstrated to be the site of blood–brain barrier disruption in this model[11]. Slides were washed with PBS and fixed in 4% paraformaldehyde for 15 min at room temperature. Samples were then washed three times with 100 μl PBS and incubated for 1 h in 5% normal goat serum and 0.5% Igepal CA-630 in a total volume of 100 μl (Sigma-Aldrich, St. Louis, MO). Slides were then incubated with rabbit anti-mouse occludin (clone OC-3F10, Invitrogen, Carlsbad, CA) at 1:200 dilution overnight at 4 °C in a total volume of 100 μl. Samples were washed three times with PBS, and stained with Alexa Fluor 647 Goat anti-rabbit IgG secondary antibody in a total volume of 100 μl (Invitrogen, Carlsbad, CA, 1:500 dilution) for 1 h. All slides were washed five times with 100 μl PBS. Slides were dried and covered with Vectashield medium with DAPI (Vector lab, Burlingame, CA). Images were acquired using a Leica (Wetzlar, Germany) DM2500 equipped with a ×63 oil immersion objective. All images were collected at room temperature and analyzed using Leica Acquisition Suite software. Samples were additionally analyzed using ImageJ software (NIH, Bethesda, MA).

**FITC-albumin permeability assay**. The left hemisphere of FITC-albumin injected brains were homogenized in radioimmunoprecipitation assay (RIPA) buffer (10 mmol/l Tris, 140 mmol/l NaCl, 1% Triton X-100, 1% Na dexycholate, 0.1% SDS and protease inhibitor cocktail pH 7.5) using a PowerGen 125 homogenizer (Fisher Scientific, Hampton, NH). Samples were then centrifuged for 10 min at 10,000 rpm at 4 °C and supernatants were saved. Protein concentration in the supernatant was determined using a BCA protein assay (Pierce Biotechnology, Waltham, MA) and read on a Synergy H1 Hybrid Multi-Mode Reader (BioTek, Winooski, VT). Samples were normalized for protein content, and read on a fluorescent plate reader at 488 nm excitation and 525 nm emission.

**Acute viral infection and vaccination**. The recombinant TMEV XhoI-OVA8 (TMEV-OVA) strain was generated by our group[29]. Eight to 12-week-old animals

infected intracranially (i.c.) with TMEV-OVA were anesthetized with 1–2% iso-flurane, then received a single dose of 2×10$^5$ plaque forming units (PFU) of TMEV in the right hemisphere of the brain in a total volume of 10 μl. GL261-quad cassette-bearing animals were vaccinated in the left hemisphere, the opposite hemisphere from where the GL261-quad cassette cells were implanted.

**GL261 cell culture and implantation**. The GL261-quad cassette cell line was provided by the lab of John R. Ohlfest, PhD from the Masonic Cancer Center (University of Minnesota, Minneapolis, MN). The GL261-quad cassette cell line is engineered to express four model antigens: the K$^b$-restricted OVA$_{257-264}$, as well as OVA$_{323-339}$, human GP100$_{25-33}$, and the alloantigen I-E$^a_{52-68}$[5–7]. The GL261-quad cassette cell line also expresses luciferase, which serves as a surrogate for tumor size via bioluminescence imaging. 6×10$^4$ GL261-quad cassette cells were implanted by stereotactic injection into the right striatum of each animal[5–7]. Prior to implantation, 8- to 12-week-old animals were anesthetized with 20 mg/kg ketamine and 5 mg/kg xylazine to minimize discomfort during the procedure. Each mouse was positioned on a stereotactic injection platform. Cells were injected in a total volume of 1 μl at a rate of 0.2 μl per minute and the site of injection was 1 mm lateral, 2 mm anterior of the bregma. The injection depth was 3 mm from the surface of the cortex. Animals met endpoint criteria when they lost 20% of their starting body weight. All animal experiments were approved by and performed in accordance with the Mayo Clinic Institutional Animal Care and Use Committee.

**Bioluminescence imaging and analysis**. GL261-quad cassette-bearing animals were assessed for tumor burden using bioluminescence imaging[5–7]. Animals received 150 mg/kg D-luciferin in PBS in a total volume of 200 μl intraperitoneally (i.p.) (Gold Biotechnology, Olivette, MO). Animals were subsequently anesthetized with 1–2% isoflurane before and during imaging. Animals were imaged using an IVIS Spectrum system (Xenogen Corp., Amameda, CA, USA) running Living Image software. Bioluminescence intensity (photons/sec) was recorded in a circular region of interest surrounding the head over a 10-s period. All animal work was completed in accordance to the Mayo Clinic Institutional Animal Care and Use Committee guidelines.

**Flow cytometry**. Spleens and thymuses were harvested in 5 ml RPMI and pressed through a 70 μm filter to achieve a single cell suspension. Brains were harvested in 5 ml RPMI and manually homogenized using a dounce homogenizer[49]. Brain samples were then filtered through a 70 μm filter into a 50% percoll solution for a total volume of 25 ml. Samples were centrifuged at 7840×g and the myelin debris layer was removed. All samples were washed twice and plated in a 96-well v bottom plate. Peptide:MHC tetramers were constructed by our group and samples were stained at a 1:100 dilution of tetramer for 30 min on ice in the dark[5]. Antibodies against CD45 (violetFluor 450-conjugated, clone 30F-11, Tonbo Biosciences, San Diego, CA, 1:100 dilution), CD4 (PerCP Cy5.5-conjugated, clone RM4-5, Tonbo Biosciences, San Diego, CA, 1:100 dilution), CD11c (Brilliant Violet 650-conjugated, clone HL-3, BD Biosciences, San Jose, CA, 1:100 dilution), CD11b (APC-conjugated, clone M1/70, BD Biosciences, San Jose, CA, 1:100 dilution), GR-1 (FITC-conjugated, clone R86-8C5, BD Biosciences, San Jose, CA, 1:100 dilution), and H-2K$^b$ (PE-conjugated, clone AF6-88.5, BD Biosciences, San Jose, CA, 1:100 dilution) were used for staining at a 1:100 dilution in PBS for 45 min on ice in the dark in addition to a 1:1000 dilution of Ghost Red 780 Viability Dye (Tonbo Biosciences, San Diego, CA). Anti-CD8α antibody (Brilliant Violet 785-conjugated, clone 53–6.7, BioLegend, San Diego, CA, 1:100 dilution) was added during the final 15 min of the staining period at a 1:100 dilution. To assess TCR Vβ usage, samples were stained using an internal pre-diluted TCR Vβ antibodies (Mouse V β TCR Screening Panel, pre-diluted FITC-conjugated monoclonal antibodies, BD Biosciences, San Jose, CA) in addition to antibodies against CD45 (violetFluor 450-conjugated, clone 30F-11, Tonbo Biosciences, San Diego, CA, 1:200 dilution), CD4 (PerCP Cy5.5-conjugated, clone RM4-5, Tonbo Biosciences, San Diego, CA, 1:200 dilution), and CD8α (Brilliant Violet 785-conjugated, clone 53–6.7, BioLegend, San Diego, CA, 1:200 dilution) for 30 min. Samples were fixed with 2% PFA and run on a BD LSRII flow cytometer equipped with FACSDiva software (BD Biosciences, San Jose, CA). Additionally, samples were digitally compensated using single-stained controls and analyzed by FlowJo v10 software (FlowJo LLC, Ashland, OR). All gating schemes may be found in Supplementary Figure 1.

**Statistical analysis**. All data are presented as mean ± standard error of the mean (SEM). Significance was determined using a two-sided Student's t test, a one-way ANOVA with Holm–Sidak correction for multiple comparisons, or a two-way ANOVA with Holm–Sidak correction for multiple comparisons. A Mann–Whitney Rank Sum Test was used if data did not follow a normal distribution. Significant differences in survival were determined using a logrank (Mantel–Cox) test. GraphPad Prism 7.0 (La Jolla, CA) was used for all statistical analysis.

**Data availability**. The data that support the findings of this study are available from the corresponding author upon request.

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

## Acknowledgements

We would like to acknowledge Larry R. Pease, Ph.D., for sharing of the H-2K$^b$ transgene used in this system. We would like to thank John T. Harty, Ph.D. for providing the *Plasmodium berghei* ANKA parasite. The authors received funding for this work through the National Cancer Institute (R21 CA186976), The National Institute of Neurologic Disease and Stroke (R21 NS094765, R56 NS094765, and R01 NS103212), the Mayo Clinic-Koch Institute Collaboration, and the Mayo Foundation for Medical Education and Research.

## Author contributions

C.S.M., M.A.H., E.N.G, D.N.R., L.R.P., K.D.P., and A.J.J. conceived the experiments and developed the methodology. C.S.M., M.A.H., E.N.G., H.M.A.T., D.N.R., F.J., and M.J.H.

performed the experiments. K.D.P. and L.R.P. provided resources. C.S.M. completed formal analysis and wrote the manuscript. All authors reviewed and commented on the manuscript. A.J.J. acquired funding and managed the project.

## Additional information

**Competing interests:** The authors declare no competing financial interests.

