## [Peer Review File · Nature Communications]

Reviewers' comments:

Reviewer #1 (Remarks to the Author):

In this interesting paper, the authors use three different models of neuroinflammation to study the differential role of macrophages and dendritic cells to activate CD8 T cell responses. They conclude that dendritic cells and macrophages have distinct roles in neuroinflammation induced by picornavirus infection, experimental cerebral malaria, and syngeneic glioma. An elegant approach is used to define antigen presentation function of specific APCs. They generated a novel transgenic mouse model that enables conditional deletion of the H-2Kb (Kb) MHC class I molecule using a cre-lox system in macrophages or dendritic cells. This is an innovative approach.

The most interesting data is that CD11c-cre Kb cKO and CMV-cre Kb cKO animals maintain organization of blood vessels in the CNS and CD11c-cre KbcKO animals were fully protected from ECM with 100% of animals surviving. This offers a novel approach for cerebral malaria therapies. Although the overall conclusion that dendritic cells are superior to macrophages in inducing CD8 responses is not surprising, the paper presents new data in differentiating between APCs in the context of CD8-mediated neuroinflammation. Overall, the manuscript is clearly presented and well-written. In spite of the list of open questions and future work that this paper stimulates (some of these are listed by the authors), the manuscript presents novel information. One potential drawback to the manuscript as presented, is the lack of clear mechanistic explanation. Due to the application of multiple, parasite, viral and tumor models, it is unclear how CD11c or LysM expressing cells contribute to the individual disease. What is so specific about the malaria-induced CD8 T cells in destroying BBB in ECM but not in other models? It would have been interesting to analyze in more details the nature of CNS infiltrating CD8 T cells. Finally, the caveats of CD11c and LysM specific models should be discussed. Since the work solely relies on the application of CD11c and LysM promoters, the potential of CD11c or LysM expression by other cell types should be considered. Deletion of Kb in these different cell types might happen with different efficiency, particularly if we consider different tissue specific deletion efficiency. MFI shown in Figure 1f-g might not adequately reflect deletion efficiency.

Some specific concerns are noted:

1. While CD11c promoter and LysM promoter are useful for defining CD11c+ and LysM-expressing cell subtypes, the "leakage" of these promoters to other cell types should be discussed. For example, activated microglia might express CD11c in neuroinflammation, thus local restimulation of infiltrating cells that could contribute to CD8 cell survival and detection in the CNS by flow. CD11c expressing cells differ in different tissues, and it is unclear whether Kb deletion would be equal in peripheral or CNS tissues. Similarly, LysM expressing cell composition might be different in tissues or circulation. LysM driven deletion affects neutrophils also and this should be measured.
2. Due to the low percentage differences shown in Figure 3d-f, absolute number of infiltrating CD8 and CD4 cells in CD11c-cre and LysM-cre Kb cKO animals six days post infection should be shown.
3. Results suggest a requirement for Kb -restricted antigen presentation by both macrophages and dendritic cells, however, it is not clear whether this would take place in the periphery or locally in the CNS. This concern should be addressed in more details.
4. The overall mechanism is a bit confusing. They show that there were no differences in circulating parasite load between CD11c-cre Kb cKO, LysM-cre Kb cKO, and cre-negative animals, however the disease course was different. While, LysM-cre Kb cKO, and cre-negative animals become moribund and succumb to disease CD11c-cre Kb cKO and CMVCre Kb KO animals survived and their BBB was intact. What is happening with the circulating parasites? Other mechanisms excluding CD8 T cells for BBB breach had been proposed previously. How could this be integrated with these present results?
5. Figure 4b should indicate exact brain areas for these images. Additionally, occluding staining for CD11cKO is difficult to interpret. Either a better quality image or higher magnification should be included.
6. Figure 4 title should be revised. CMV-cre Kb cKO animals are also protected.

7. They indicated that commercially available CD11c-cre, LysM-cre, and CMV-cre animals were used, however the exact strain code, stock number and donating source references were not included. Due to the differences between different Cre transgenics, exact strain specifications should be listed.

Reviewer #2 (Remarks to the Author):

This is a fine report that addresses an important question. The authors have thoughtfully developed excellent tools and have used them well. The experiments are well controlled and the conclusions appropriate. The problem is, and this would surely disappoint the authors, that the conclusions really support the hypothesis is that the CNS is not much different from other tissues. The DCs really are the key players in priming immune responses, with macrophage playing little or no role in the priming process itself even as they do activate CD8 T cells. The authors try to massage this point by saying that "dendritic cells and macrophages both activate CD8 T cell responses... the extent to which each of these APCs contributed to CD8 T cell priming varied." Nonetheless, their carefully derived data are clear that as elsewhere, DCs are the main drivers of priming. Lastly, some key literature (such as Walter L, Albert ML. J Immunol. 2007 May 15; 178: 6038-42) is not cited.

Reviewer #3 (Remarks to the Author):

Here Malo and her colleagues provide potential new insights into the role of MHC expressing antigen presenting cell types in diseases of the CNS. They generated mice in which one the leader exon of the Kb transgene was flanked by loxp sites. In mice devoid of endogenous Kb or Db MHC, this transgene became the only source of Kb MHC I and its expression could be specifically deleted in cells expressing the Cre recombinase. Using mice expressing CMV-Cre, LysM-Cre or CD11c-Cre they were able to generate mice without any Kb (CMV-Cre, ubiquitous Cre), or mice in which Kb was lacking only in macrophages (LysM-Cre, macrophage specific Cre) or dendritic cells (CD11c-Cre, DC specific Cre) respectively. The authors noted that CD8+ T cell development in the thymus and spleen appeared to be normal. These three mouse strains were then used to assess the requirement for Kb expressing macrophages or DC in three different diseases of the CNS.

In the first model of experimental cerebral malaria, the infiltration of CD8+ T cells in the CNS and subsequent morbidity was completely abrogated in mice lacking Kb as well as in mice deficient in Kb expressing DCs or macrophages. By contrast, in a Theiler's virus infection or in a glioma model the CD8+ T cell responses to a viral or glioma antigen (OVA), the DCs were critical to elicit OVA-specific CD8 T cell responses as well as control the virus or tumor growth. Thus the main and not too surprising conclusion is that antigen presentation by DCs is key to generation of CD8+ T cell responses.

My concerns with this study are that first, the authors seem to take the cell numbers in the thymus as proving the redundancy of DC and macrophages in T cell development. Given previous literature on the specific role of DCs and macrophages in negative and positive selection stages of T cell development, it is likely that the T-cell repertoires are quite different in these mice, but cannot be judged by a simple cell count. Perhaps if the authors had tested a panel of model or allo-antigens they would have a better sense of this important point that impinges on all their subsequent experiments. Of course it is also good to remember that the non-classical MHC class I molecules are intact in these animals and do contribute to the CD8 T cell development and repertoire.

Altogether, I am impressed by the elegant approach. If the T-cell repertoires in these animals are

indeed identical these nice experiments have revealed apparently non-redundant roles for DC and macrophages in these disease models.

Reviewer Reports:

Reviewer #1:

1. While CD11c promoter and LysM promoter are useful for defining CD11c⁺ and LysM-expressing cell subtypes, the “leakage” of these promoters to other cell types should be discussed. For example, activated microglia might express CD11c in neuroinflammation, thus local restimulation of infiltrating cells that could contribute to CD8 cell survival and detection in the CNS by flow. CD11c expressing cells differ in different tissues, and it is unclear whether Kb deletion would be equal in peripheral or CNS tissues. Similarly, LysM expressing cell composition might be different in tissues or circulation. LysM driven deletion affects neutrophils also and this should be measured. **To be responsive to this review, we have incorporated an additional supplemental figure (Supplemental Figure 2 and Page 5, Lines 93-94). In this supplemental figure, we provide data that demonstrates that there is only very limited deletion of H-2K^b on activated CD11c⁺ microglia. The vast majority of microglia retain expression of H-2K^b (Page 5, Lines 93-94).**

Interestingly, we do observe deletion of H-2K^b on neutrophils (CD11b⁺GR-1⁺ cells). A major reason that our emphasis remains on macrophages, however, is that this immune cell type has historically been defined as a “professional antigen presenting cell”. We therefore focused on this cell type’s capacity to be a primary driver of antigen presentation.

2. Due to the low percentage differences shown in Figure 3d-f, absolute number of infiltrating CD8 and CD4 cells in CD11c-cre and LysM-cre Kb cKO animals six days post infection should be shown. **Obtaining a completely accurate count of brain infiltrating lymphocytes is not feasible given the extent solid tissues need to be processed to liberate immune cells. Nevertheless, we understand the reviewer’s request to analyze the magnitude of immune cell entry into the brain. To address this, we have reanalyzed our data to incorporate cell numbers as fraction of total events analyzed by flow cytometry. This way we take into account cell infiltration into the brain in addition to determining frequency of cells as a percentage of blood-derived (CD45 high) cells analyzed. Our analysis of cell number per 100,000 TOTAL events has now been incorporated into Figure 4.**

3. Results suggest a requirement for Kb -restricted antigen presentation by both macrophages and dendritic cells, however, it is not clear whether this would take place in the periphery or locally in the CNS. This concern should be addressed in more details. **We agree this is an interesting and important question that merits further study. We also intend to use our novel transgenic mouse system to address this question. However, as it stands, our system was carefully designed to only incorporate one MHC class I molecule to reduce the effect of an additional T cell response specific to unrelated antigen presentation. While this approach reduced background from competing MHC class I, our strategy makes us unable to use classical adoptive transfer approaches to analyze the location of CD8 T cell proliferation and expansion (MHC class I will not be matched). Therefore, while we intend to address this important question, conducting this experiment appropriately will require additional breeding and further developing our K^b LoxP conditional knockout mouse system. Therefore, these studies are important but will be conducted as a separate future line of investigation.**

4. The overall mechanism is a bit confusing. They show that there were no differences in circulating parasite load between CD11c-cre Kb cKO, LysM-cre Kb cKO, and cre-negative animals, however the disease course was different. While, LysM-cre Kb cKO, and cre-negative animals become moribund and succumb to disease CD11c-cre Kb cKO and CMVCre Kb KO animals survived and their BBB was intact. What is happening with the circulating parasites? Other mechanisms excluding CD8 T cells for BBB breach had been proposed previously. How could this be integrated with these present results? **We have clarified this in this manuscript. Parasite load alone does not result in changes in blood-brain barrier disruption. Rather, it is the CD8 T cell responses against the parasite that promotes this neuropathology. Our group, as well as others, has demonstrated that blood-brain barrier disruption is exclusively mediated by perforin expressing CD8 T cells in the *Plasmodium berghei* ANKA model of experimental cerebral malaria (PMID #s 6749988, 8759747, 9011069, 12574396, 16500656, 21525386, 28264905). While some studies have suggested a role for other factors, these findings were primarily conducted in *in vitro* blood-brain barrier systems. Additionally, we are not aware of *in vivo* studies in which CD8 T cells are not the mediator of blood-brain barrier disruption during acute PBA infection. There are reports using various knockout mouse strategies that demonstrate resistance to blood-brain barrier disruption. However, these knockout mice have attenuated CD8 T cell responses, which explains their resistance to blood-brain barrier disruption. Pertaining to our studies, CMV-cre and CD11c-cre K^b cKO mice survive acute PbA infection because there is no CD8 T cell-mediated blood-brain barrier disruption. However, these animals will eventually succumb to anemia due to increased parasite load during chronic stages of disease at three weeks post infection.**

5. Figure 4b should indicate exact brain areas for these images. Additionally, occluding staining for CD11cKO is difficult to interpret. Either a better quality image or higher magnification should be included. **We have replaced the original image with one of higher resolution (Figure 5). We have also edited the methods section to include the region of the brain imaged (Page 36, Lines 590-591).**

6. Figure 4 title should be revised. CMV-cre Kb cKO animals are also protected. **This has been modified as part of our resubmission (Page 15, Line 217).**

7. They indicated that commercially available CD11c-cre, LysM-cre, and CMV-cre animals were used, however the exact strain code, stock number and donating source references were not included. Due to the differences between different Cre transgenics, exact strain specifications should be listed. **This has been incorporated as part of our resubmission (Page 35, Lines 570-573).**

Reviewer #2 Specific Concerns:

1. The conclusions support the hypothesis that the CNS is similar to other tissues. **We agree that, during acute virus infection, the conclusion is that the CNS is not drastically different from other tissues. However, we demonstrate a nonredundant role for these APCs in experimental cerebral malaria and vaccination against glioma. Nevertheless, we contend the brain having some similarities to peripheral tissue is an important finding, as the CNS has historically been considered immunologically distinct. As a result, this belief has been very detrimental to the development of immunotherapy for neurologic disease. The most striking example being in the field of neuro-oncology, where immunotherapy strategies and clinical trials for CNS tumors have greatly lagged behind work conducted in peripheral cancers.**

2. Nonetheless, their carefully derived data are clear that as elsewhere, DCs are the main drivers of priming. Lastly, some key literature (such as Walter L, Albert ML. J Immunol. 2007 May 15;178:6038-42) is not cited. **To**

be responsive to this review, we have cited the relevant literature as thoughtfully suggested by reviewer 2 (Page 4, Lines 64-65).

Reviewer #3 Specific Concerns:

1. Given previous literature on the specific role of DCs and macrophages in negative and positive selection stages of T cell development, it is likely that the T-cell repertoires are quite different in these mice, but cannot be judged by a simple cell count. **We agree that not knowing the TCR repertoire could impact conclusions drawn from our studies. To be responsive to this thoughtful review, we have included flow cytometric analysis of TCR V β usage in each of our transgenic animals, which has been incorporated as a new figure (Figure 3). We assessed TCR V β usage in splenic CD4 and CD8 T cells. We found only very minor differences in TCR V β 5 proportion amongst splenic CD8 T cells. Additionally, the CD4 T cell compartment remained unchanged in CD11c-cre K^b cKO and LysM-cre K^b cKO mice when compared to controls. Therefore, we contend that the T cell repertoire is not significantly altered in our animals where class I molecules are conditionally deleted on APC subsets. In addition, we assessed TCR V β usage in the thymus at the DN, DP, CD4SP and CD8SP stages of development (Figure S3). Similar to our findings in splenic T cells, we demonstrate minimal changes in the repertoire of CD8 and CD4 single positive thymocytes.**

REVIEWERS' COMMENTS:

Reviewer #1 (Remarks to the Author):

The paper is significantly improved due to the incorporation of additional supplementary data and reanalysis of the infiltrating cell numbers as fraction of the total events in Fig 4. The exact localization of CD8 T cell induction is still a question, however, this would require substantial further work. Overall, the manuscript makes an important point regarding discrete roles of macrophages and dendritic cells in inducing CNS infiltrating CD8 T cells.

Reviewer #3 (Remarks to the Author):

In this revised version of their original manuscript, Malo and her colleagues provide a well-reasoned rebuttal to the criticisms and comments of the previous reviewers. They have carried out additional experiments to address key issues such as the potential differences in the repertoire of T-cell antigen receptors (ab TCRs). At the first approximation, based upon the analysis of Vb usage, it appears that there are minimal, if any, differences among the TCRs usage in the spleen as well as the thymuses of the different mouse models. This analysis makes it less likely that highly skewed TCR repertoires could account for their observations in the disease models. Although it is theoretically possible that changes in the usage of TCR alpha chains or in the junctional nucleotides that comprise the complementarity determining regions (CDRs) of the TCRs could alter the T-cell responses, such analysis of TCR sequences is clearly outside the scope of this work.

In sum, I agree that the authors have satisfactorily demonstrated a surprisingly distinct non-redundant role for macrophages and dendritic cells in different CNS diseases.

We appreciate the careful review of our manuscript entitled “Non-equivalent antigen presenting capabilities of dendritic cells and macrophages in generating brain-infiltrating CD8⁺ T cell responses” by *Nature Communications*. We have included the remarks from reviewers below.

Reviewer Reports:

Reviewer #1 (Remarks to the Author):

The paper is significantly improved due to the incorporation of additional supplementary data and reanalysis of the infiltrating cell numbers as fraction of the total events in Fig 4. The exact localization of CD8 T cell induction is still a question, however, this would require substantial further work. Overall, the manuscript makes an important point regarding discrete roles of macrophages and dendritic cells in inducing CNS infiltrating CD8 T cells.

We agree with Reviewer #1 and thank them for their careful review of this manuscript. We agree that there are exciting future directions for this work.

Reviewer #3 (Remarks to the Author):

In this revised version of their original manuscript, Malo and her colleagues provide a well-reasoned rebuttal to the criticisms and comments of the previous reviewers. They have carried out additional experiments to address key issues such as the potential differences in the repertoire of T-cell antigen receptors (ab TCRs). At the first approximation, based upon the analysis of Vb usage, it appears that there are minimal, if any, differences among the TCRs usage in the spleen as well as the thymuses of the different mouse models. This analysis makes is less likely that highly skewed TCR repertoires could account for their observations in the disease models. Although it is theoretically possible that changes in the usage of TCR alpha chains or in the junctional nucleotides that comprise the complementarity determining regions (CDRs) of the TCRs could alter the T-cell responses, such analysis of TCR sequences is clearly outside the scope of this work.

In sum, I agree that the authors have satisfactorily demonstrated a surprisingly distinct non-redundant role for macrophages and dendritic cells in different CNS diseases.

We thank Reviewer #3 for their review of our manuscript. We agree that assessing TCR alpha chains and other regions of the TCR would be interesting, and is a point of future direction.